# Transmission of community- and hospital-acquired SARS-CoV-2 in hospital settings in the UK: A cohort study

Yin Mo [1,2,3,4]*, David W. Eyre [5,6,7,8], Sheila F. Lumley [6], Timothy M. Walker [1,6,10], Robert H. Shaw [6], Denise O'Donnell [6], Lisa Butcher [6], Katie Jeffery [6,9], Christl A. Donnelly [11,12], Oxford COVID infection review team[¶], Ben S. Cooper [1,2]

1 Oxford Centre for Global Health Research, Nuffield Department of Medicine, University of Oxford, Oxford, United Kingdom, 2 Mahidol-Oxford Tropical Medicine Research Unit, Faculty of Tropical Medicine, Mahidol University, Bangkok, Thailand, 3 Division of Infectious Disease, Department of Medicine, National University Hospital, Singapore, 4 Department of Medicine, National University of Singapore, Singapore, 5 Big Data Institute, Nuffield Department of Population Health, University of Oxford, Oxford, United Kingdom, 6 Oxford University Hospitals, NHS Foundation Trust, Oxford, United Kingdom, 7 NIHR Oxford Biomedical Research Centre, University of Oxford, Oxford, United Kingdom, 8 NIHR Health Protection Research Unit in Healthcare Associated Infections and Antimicrobial Resistance at University of Oxford in partnership with Public Health England, Oxford, United Kingdom, 9 Radcliffe Department of Medicine, University of Oxford, Oxford, United Kingdom, 10 Oxford University Clinical Research Unit, Ho Chi Minh City, Vietnam, 11 Department of Statistics, University of Oxford, Oxford, United Kingdom, 12 MRC Centre for Global Infectious Disease Analysis, Department of Infectious Disease Epidemiology, Imperial College London, London, United Kingdom

¶ Membership of Oxford COVID infection review team is provided in the Acknowledgements and in S5 Text.
* moyin@tropmedres.ac

**Data Availability Statement:** The datasets analysed during the current study are not publicly available as they contain personal data but are

## Abstract

### Background

Nosocomial spread of Severe Acute Respiratory Syndrome Coronavirus 2 (SARS-CoV-2) has been widely reported, but the transmission pathways among patients and healthcare workers (HCWs) are unclear. Identifying the risk factors and drivers for these nosocomial transmissions is critical for infection prevention and control interventions. The main aim of our study was to quantify the relative importance of different transmission pathways of SARS-CoV-2 in the hospital setting.

### Methods and findings

This is an observational cohort study using data from 4 teaching hospitals in Oxfordshire, United Kingdom, from January to October 2020. Associations between infectious SARS-CoV-2 individuals and infection risk were quantified using logistic, generalised additive and linear mixed models. Cases were classified as community- or hospital-acquired using likely incubation periods of 3 to 7 days. Of 66,184 patients who were hospitalised during the study period, 920 had a positive SARS-CoV-2 PCR test within the same period (1.4%). The mean age was 67.9 (±20.7) years, 49.2% were females, and 68.5% were from the white ethnic group. Out of these, 571 patients had their first positive PCR tests while hospitalised (62.1%), and 97 of these occurred at least 7 days after admission (10.5%). Among the 5,596 HCWs, 615 (11.0%) tested positive during the study period using PCR or serological

available from the Infections in Oxfordshire Research Database (https://oxfordbrc.nihr.ac.uk/research-themes-overview/antimicrobial-resistance-and-modernising-microbiology/infections-in-oxfordshire-research-database-iord/), subject to an application and research proposal meeting the ethical and governance requirements of the Database. All analysis codes are available at https://github.com/moyinNUHS/covid_HospTransmissionDynamics.

**Funding:** YM is supported by the Singapore National Medical Research Council Research Fellowship (NMRC/Fellowship/0051/2017). BSC acknowledges support from the Medical Research Council (MRC) (MR/V028456/1). TMW is a Wellcome Trust Clinical Career Development Fellow (214560/Z/18/Z). CAD acknowledges funding from the MRC Centre for Global Infectious Disease Analysis (MR/R015600/1), jointly funded by the UK MRC and the UK Foreign, Commonwealth & Development Office (FCDO), under the MRC/FCDO Concordat agreement and is also part of the EDCTP2 programme supported by the European Union, Vaccine Efficacy Evaluation for Priority Emerging Diseases (VEEPED) grant (NIHR: PR-OD-1017–20002) from the NIHR. This work was also supported by the NIHR HPRU in Healthcare Associated Infections and Antimicrobial Resistance at Oxford University in partnership with PHE (NIHR200915), the NIHR Biomedical Research Centre, Oxford, and the NIHR HPRU in Emerging and Zoonotic Infections at University of Liverpool in partnership with PHE, in collaboration with Liverpool School of Tropical Medicine and the University of Oxford (NIHR200907). The Mahidol-Oxford Tropical Medicine Research Unit received funding by the Wellcome Trust [220211]. The funders had no role in study design, data collection and analysis, decision to publish, or preparation of the manuscript.

**Competing interests:** I have read the journal's policy and the authors of this manuscript have the following competing interests: DWE declares personal fees from Gilead outside the submitted work.

**Abbreviations:** aOR, adjusted odds ratio; COVID-19, Coronavirus Disease 2019; CrI, credible interval; HCW, healthcare worker; PPE, personal protective equipment; SARS-CoV-2, Severe Acute Respiratory Syndrome Coronavirus 2; WAIC, Widely Applicable Information Criterion.

tests. The mean age was 39.5 (±11.1) years, 78.9% were females, and 49.8% were nurses. For susceptible patients, 1 day in the same ward with another patient with hospital-acquired SARS-CoV-2 was associated with an additional 7.5 infections per 1,000 susceptible patients (95% credible interval (CrI) 5.5 to 9.5/1,000 susceptible patients/day) per day. Exposure to an infectious patient with community-acquired Coronavirus Disease 2019 (COVID-19) or to an infectious HCW was associated with substantially lower infection risks (2.0/1,000 susceptible patients/day, 95% CrI 1.6 to 2.2). As for HCW infections, exposure to an infectious patient with hospital-acquired SARS-CoV-2 or to an infectious HCW were both associated with an additional 0.8 infection per 1,000 susceptible HCWs per day (95% CrI 0.3 to 1.6 and 0.6 to 1.0, respectively). Exposure to an infectious patient with community-acquired SARS-CoV-2 was associated with less than half this risk (0.2/1,000 susceptible HCWs/day, 95% CrI 0.2 to 0.2). These assumptions were tested in sensitivity analysis, which showed broadly similar results. The main limitations were that the symptom onset dates and HCW absence days were not available.

## Conclusions

In this study, we observed that exposure to patients with hospital-acquired SARS-CoV-2 is associated with a substantial infection risk to both HCWs and other hospitalised patients. Infection control measures to limit nosocomial transmission must be optimised to protect both staff and patients from SARS-CoV-2 infection.

### Author summary

#### Why was this study done?

- Transmission of Severe Acute Respiratory Syndrome Coronavirus 2 (SARS-CoV-2) in the hospital setting has been widely reported, but little is known about the incidence and pathways of transmission.

- Hospitalised patients are especially vulnerable to Coronavirus Disease 2019 (COVID-19)-associated complications, and infected patients may contribute to the further spread of SARS-CoV-2 in the community and nursing homes upon discharge.

- Healthcare workers (HCWs) are disproportionately infected with SARS-CoV-2, and a reduced staff workforce due to SARS-CoV-2 infection may compromise the clinical management of patients and infection prevention and control measures.

- Improved understanding of the drivers of hospital-acquired SARS-CoV2 infection is important to prevent and control the spread of SARS-CoV-2 in hospitals.

#### What did the researchers do and find?

- We collected data from 4 teaching hospitals in Oxfordshire, United Kingdom, from January to October 2020.

- The data were analysed to find the associations between infectious SARS-CoV-2 individuals (classified as community- or hospital-acquired) and infection risk posed to the susceptible individuals using statistical models.

- For susceptible patients, 1 day in the same ward with another patient with hospital-acquired SARS-CoV-2 was associated with an additional 8 infections per 1,000 susceptible patients, while exposure to an infectious patient with community-acquired COVID-19 or to an infectious HCW was associated with substantially lower infection risks of 2 per 1,000 susceptible patients.

- As for HCW infections, exposure to an infectious patient with hospital-acquired SARS-CoV-2 or to an infectious HCW were both associated with an additional 1 infection per 1,000 susceptible HCWs per day, while exposure to an infectious patient with community-acquired SARS-CoV-2 was associated with less than half this risk.

## What do these findings mean?

- Our data provide strong evidence that newly infected COVID-19 patients are associated with a high risk of onward transmission to patients and HCWs in hospital.

- Our findings support enhanced strategies to prevent and identify early hospital-onset SARS-CoV-2 infection among hospitalised patients, for example, regular screening and prompt testing to identify these patients.

- Measures to ensure infected staff are not at work, including regular staff screening and adequate sick pay arrangements, are vital.

- The relatively low risk of transmission associated with patients with suspected community-acquired COVID-19 suggests that for these patients, the peak of their infectivity may have passed such that existing infection prevention and control policies including universal use of personal protective equipment, prompt testing, and isolation of suspected or known cases are sufficient to mitigate most of the remaining infectiousness.

- The main limitations were that the symptom onset dates and HCW absence days were not available, which may affect the estimation of the transmission pathways.

## Introduction

Nosocomial transmission and outbreaks of Severe Acute Respiratory Syndrome Coronavirus 2 (SARS-CoV-2) have been frequently reported in various healthcare settings since the beginning of the pandemic [1–6]. Reported proportions of hospitalised Coronavirus Disease 2019 (COVID-19) patients suspected to have acquired SARS-CoV-2 in the hospitals vary widely, ranging from <1% to 20% [7–10], and a national data linkage study in England estimated that 15% of laboratory-confirmed cases among hospital patients were healthcare-associated [11].

Nosocomial transmission of SARS-CoV-2 is of considerable concern. Hospitalised patients are especially vulnerable to COVID-19-associated complications and mortality [2]. Infected patients who are asymptomatic or become symptomatic after discharge from the hospital may

contribute to the further spread of SARSCoV-2 in the community and nursing homes. Health-care workers (HCWs) are disproportionately infected with SARS-CoV-2 [12–15]. They may be a key source of viral transmission to patients and fellow colleagues. Reduced staff workforce due to SARS-CoV-2 infection may compromise the clinical management of patients and infection prevention and control measures. These threats remain relevant despite the introduction of vaccines as novel variants can reduce the protection afforded, and their efficacy preventing onward transmissions may only be partial.

Analysis of detailed individual-level data including information on patients at risk of becoming infected has been lacking, and the relative importance of different transmission pathways (e.g., patient to HCW, HCW to patient, HCW to HCW, and patient to patient) has not, to our knowledge, previously been quantified [16]. Improved understanding of the drivers of nosocomial SARS-CoV2 infection is of potential value for improving targeting of infection prevention and control activities in hospitals.

The objectives of this analysis are to use high-resolution individual-level data to quantify associations between patient characteristics and risks for acquiring nosocomial SARS-CoV-2 infection after adjusting for exposures, describe how risk of acquisition changes both with calendar time and over a patient's hospital stay, and provide evidence about the relative importance of different transmission pathways for both patients and HCWs.

## Methods

### Study cohort

Data were obtained from Oxford University Hospitals, a group of 4 teaching hospitals (denoted hospital A to D) in Oxfordshire, United Kingdom, from 12 January 2020 to 2 October 2020. Of the 4 hospital sites, 2 (hospitals A and C) have an emergency department and admitted symptomatic SARS-CoV-2 patients directly from the community. Patient data included patient demographics, location in the hospital on every day of stay, total length of stay, and SARS-CoV-2 PCR test results (S4 Text).

SARS-CoV-2 infections in hospital HCWs were identified using PCR results from symptomatic and asymptomatic testing at the hospital. Symptomatic testing was offered to staff from 27 March 2020 onwards, and staff could participate in a voluntary asymptomatic screening programme from 23 April 2020 onwards, offering testing up to once every 2 weeks. Additionally, probable infections prior to widespread availability of testing were identified in staff without a positive PCR result, but who were either anti-nucleocapsid or anti-spike IgG antibody positive and recalled a date of onset of symptoms consistent with COVID-19. These symptoms were the presence of fever and new persistent cough, or anosmia or loss of taste [17,18].

Data were analysed taking individual infection outcomes for each person day at risk of infection as the outcome data. Thus, the dependent variable was the binary outcome coded as 1 if the individual at risk became infected on that day and coded as 0 otherwise. Independent variables were classified as time-fixed and time-varying variables. Time-fixed variables included age at admission, sex, and ethnicity routinely collected in hospital records. Time-varying variables included patients' ward and hospital location, and the number of other patients and HCWs known to be infected (and likely infectious) present on the same ward while a patient was at risk of becoming infected with SARS-CoV-2. Hospital HCWs and patients who were on the same wards on the same day were included in the analysis.

Deidentified patient data and data from HCW testing were obtained from electronic health-care records using the Infections in Oxfordshire Research Database (IORD), which has generic Research Ethics Committee, Health Research Authority and Confidentiality Advisory Group approvals (19/SC/0403, 19/CAG/0144). The study did not have a prospective protocol or

analysis plan, and the analyses were data driven. This study is reported as per the Strengthening the Reporting of Observational Studies in Epidemiology (STROBE) guideline (S1 Checklist).

### Definitions and assumptions

**Incubation period.** We assumed that each individual could only be infected once, and hence, patients and HCWs were no longer at risk for acquiring SARS-CoV-2 after their first positive PCR test. The day each patient with a potential nosocomial infection became infected is unknown, but based on knowledge of the incubation period distribution, we expect it to be 1 to 20 days prior to the date of symptom onset, with 83% falling between 3 and 7 days [19]. For a given incubation period, $d$, we assume that each patient with a nosocomial infection became infected $d$ days before the date of symptom onset.

Among 245 inpatients testing positive after developing SARS-CoV-2 symptoms during hospitalisation, the mean interval between symptom onset and their swab for PCR testing was 1 day (interquartile range 1 to 3). Consequently, we assumed that swabs for SARS-CoV-2 PCR tests after hospital admission were taken in response to COVID-19-like symptom onset 1 day earlier or, in asymptomatic cases, the swabs were assumed to have been taken 1 day after $d + 1$ days after the day of infection. The date of each patient's first positive PCR test refers to the date the swab was obtained, rather than tested if this differed (Fig 1).

**Definitions of nosocomial SARS-CoV-2 infections.** Nosocomial SARS-CoV-2 infections have previously been defined as "probable" when symptoms onset is on day 8 to 14 after admission and "definite" when symptoms onset is on day >14 after admission [20]. These increasing thresholds correspond to higher certainties that a case is hospital acquired (S2 Fig) [20]. In this study, however, we used incubation periods that are the most likely to identify the exposure risk factors, i.e., the locations and infectious individuals the susceptible individuals were exposed to, which could have resulted in an observed infection event. Our baseline

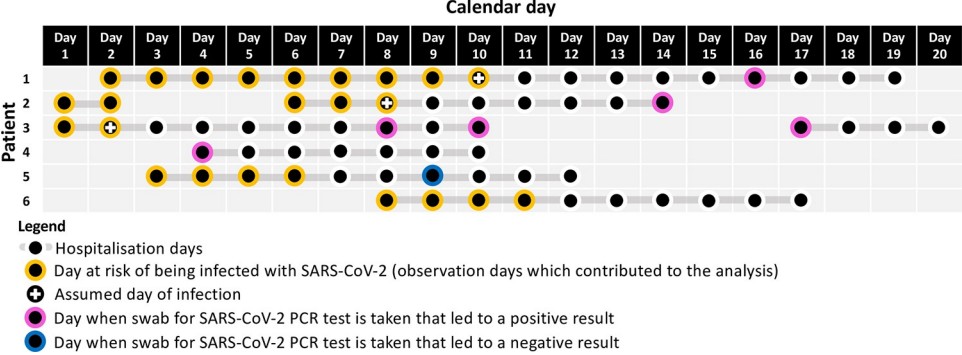

**Fig 1. Illustration of assumed incubation periods and the data analysed for 6 example patients.** We assumed that PCR tests were performed 1 day after developing symptoms, which were consistent with COVID-19. In this schematic, an incubation period of 5 days was used. Each hospitalised patient day from admission until (and including) the day of the assumed infection event (i.e., 6 (incubation period plus 1) days prior to the swab leading to the patient's first positive PCR test) or 6 days prior to the day of discharge or death (whichever occurred first) was considered an observation where the patient was at risk of becoming infected. Each observation, unique to a specific patient on a specific day, therefore corresponds to an outcome 6 days later when the patient could potentially have a swab taken for a SARS-CoV-2 PCR test. An observation had a positive outcome (value of 1) if the patient had a positive PCR test for the first time resulting from a swab taken in the hospital 6 days later, and a negative outcome (value of 0) if the patient did not have a swab taken or had a swab taken resulting in a negative PCR test 6 days later. The risk factors, e.g., ward, number of infectious patients, or HCWs in the same ward, for each observation were taken from the day of the observation itself when the corresponding patient was at risk of becoming infected. COVID-19, Coronavirus Disease 2019; HCW, healthcare worker; SARS-CoV-2, Severe Acute Respiratory Syndrome Coronavirus 2.

assumption was that the incubation period was 5 days (which is reported to be the median value) [20], and we therefore define hospital-acquired infections to be any PCR-confirmed SARS-CoV-2 infection where the patient was a hospital inpatient 6 days prior to the first positive PCR test. We also report results for sensitivity analyses assuming incubation periods of 3 and 7 days. Community-acquired infections are defined to be any PCR-confirmed infections in patients who were not hospitalised in the 20 days prior to their first positive PCR tests.

**Accounting for varying infectiousness.** We assumed that patients were infectious for a period of 10 days starting a day after the day of presumed infection, consistent with estimates that 99.7% of onward infection takes place within the first 10 days after the presumed infection event [21]. HCWs were assumed to be infectious from a day after the day of assumed infection to the day of symptom onset or 1 day prior to having a positive PCR test (i.e., staff were assumed to be absent from work after reporting symptoms consistent with SARS-CoV-2 infection or having a positive PCR test).

In the main analyses presented in the Results section, we considered infectiousness to be binary. To account for time-varying infectiousness in relation to the time of the presumed infection event, we repeated the analysis after scaling the numbers of infectious patients and HCWs in a ward on a particular day by their relative infectiousness, using the generation time distribution derived by Ferretti and colleagues [21] such that the sum of daily terms for a single infected patient or HCW who was present in the ward throughout their entire infectious period would equal one.

**Infection prevention and control measures.** The hospitals consist of a mix of side rooms and multioccupancy bays of 2 to 4 individuals. Ventilation in these wards was largely via natural ventilation with windows. Known SARS-CoV-2–infected patients were cohorted into standard rooms. There were no airborne isolation rooms. The hospitals' infection prevention and control strategies were implemented following the UK national guidelines [22]. Since the first cases of COVID-19, HCWs showing compatible symptoms were required to immediately isolate and obtain a PCR test. Patients were encouraged to wear masks at all times, especially if leaving their immediate bed area for an investigation. Visitors were generally not permitted during the pandemic period.

There were 2 major changes made to infection prevention and control measures during the study period. Prior to 1 April 2020 (phase 1), use of "level-1" personal protective equipment (PPE; apron, gloves, a surgical face mask, and optional eye protection) was recommended for contact with patients known or suspected to have COVID-19 with use of "level-2" PPE (gown, gloves, eye protection, and FFP3/N99 mask) for aerosol generating procedures. From 1 April 2020 (phase 2), in addition to the above, universal level-1 PPE was used for all patients regardless of test results or clinical suspicion of COVID-19. From 25 April 2020 (phase 3), additionally, all patients were tested for SARS-CoV-2 by PCR on admission and at weekly intervals irrespective of symptoms. Further measures were implemented subsequently from June onwards, which included universal masking and social distancing among staff, contact tracing and isolation of exposed patients and HCWs, establishment of COVID-19 dedicated areas, improved triage, and recognition of atypical symptoms in elderly patients.

## Statistical models

We first performed exploratory analyses using univariable and multivariable logistic regression models to determine associations between risk factors and SARS-CoV-2 infection for given incubation periods (Code block A in S3 Text). Independent variables in these regressions were chosen to describe the demographics of the individuals, the type of hospital wards, and infection pressures from patients and HCWs. The same set of variables were used in both

univariable and multivariable logistic regression models. In the model where the at-risk population was admitted patients, we used infectious patients with community-acquired SARS--CoV-2, infectious patients with presumed hospital-acquired SARS-CoV-2, and infectious staff on the same ward and same day as infection pressures. In the model where the susceptible population was HCWs, we also included community cases as an additional infection pressure (S3 Fig). To circumvent testing bias due to the large increase in community testing capacity during the study period, the numbers of community cases on each day were identified using the number of community-acquired SARS-CoV-2 infections admitted to the hospital in the following week [23].

To assess how well these individual demographic factors and infection pressures from infectious patients and HCWs on the same wards accounted for the nosocomial SARS-CoV-2 infections over calendar time, we used generalised additive models, which allowed for the risk of infection to depend in a nonlinear manner on the predictors (Code block B in S3 Text). The generalised additive models were implemented using the R package *mgcv* [24].

We then modelled the patients' and HCWs' daily risk of acquiring SARS-CoV-2 in the hospital using a generalised linear mixed model with an identity link (Code block C in S3 Text). This model allowed the daily probability of infection to scale linearly with infection pressure from HCWs and patients and for their effects to be additive. Because the ward setups such as number of beds, isolation facilities, staff-to-patient ratio, and infection prevention and control measures vary, we allowed the daily probability of infection (intercepts and slopes in the models) to vary by ward. The final estimates presented subsequently are the mean and 95% credible intervals (CrIs) of the mean estimates obtained for each ward. These models were implemented with JAGS (version 4 to 10), which uses Markov chain Monte Carlo to generate a sequence of dependent samples from the posterior distribution of the parameters [25].

Two versions of the models, one with interaction terms between the phases and forces of infection from patients and HCWs and one without the interaction terms, were performed. Model comparison was done using the Widely Applicable Information Criterion (WAIC) where lower values indicate improved model fit. [26]

All analysis was performed in R version 4.0.2 [27]. The corresponding analysis code for the above models can be found in S3 Text.

### Role of the funding source

The funders had no role in study design, data collection and analysis, decision to publish, or preparation of the manuscript. The views expressed in this publication are those of the authors and not necessarily those of the UK National Health Service, the National Institute for Health Research, the Department of Health or Public Health England, the Department of Health and Social Care, and other funders.

## Results

### Patient characteristics

There were 66,184 patients admitted to the 4 hospitals from 12 January to 2 October 2020, a time period that covered only the first "wave" of infection in the UK. Among these patients, 920 (920/66,184, 1.4%) had a positive SARS-CoV-2 PCR test. Out of these, 571 patients had their first positive PCR tests while hospitalised (62.1%), and 97 were continuously admitted for at least 7 days (10.5%). The patient characteristics are shown in Table 1. The patients who likely acquired SARS-CoV-2 while in hospital (assuming incubation periods of 3, 5, or 7 days) were older, had longer lengths of stays, and more readmissions compared to patients with no positive SARS-CoV-2 PCR tests.

**Table 1. Characteristics of the patients included in the analysis.**

| | | Patients testing positive for SARS-CoV-2[#] (n = 920) | | | | Patients did not test positive for SARS-CoV-2[#] (n = 65,264) |
|---|---|---|---|---|---|---|
| | | All patients tested positive (n = 920) | Hospital-acquired infection (assuming an incubation period of 3 days) (n = 133) | Hospital-acquired infection (assuming an incubation period of 5 days) (n = 130) | Hospital-acquired infection (assuming an incubation period of 7 days) (n = 120) | |
| Age (mean age in years, sd) | | 67.9 (20.7) | 75.8 (17.3) | 76.9 (16.4) | 76.6 (16.6) | 49.1 (27.3) |
| Sex (n, %) | Female | 453 (49.2%) | 70 (52.6%) | 66 (50.8%) | 65 (54.2%) | 34,887 (53.5%) |
| | Male | 467 (50.8%) | 63 (47.4%) | 64 (49.2%) | 55 (45.9%) | 30,350 (46.5%) |
| Ethnic groups (n, %) | White | 630 (68.5%) | 107 (80.5%) | 105 (80.8%) | 100 (83.3%) | 46,942 (71.9%) |
| | Non-white | 111 (12.1%) | 0 (0%) | 2 (1.5%) | 2 (1.7%) | 5,122 (7.8%) |
| | Unknown | 179 (19.5%) | 26 (19.5%) | 23 (17.7%) | 18 (15.0%) | 13,163 (20.2%) |
| Total hospitalisation days from Jan to Oct 2020 (mean, SD) | | 17.8 (22.2) | 38.6 (32.2) | 41.3 (32.5) | 42.1 (33.0) | 5.8 (11.8) |
| Total number of admissions from January to October 2020 (mean, SD) | | 1.7 (1.2) | 1.9 (1.5) | 2.0 (1.5) | 2 (1.5) | 1.4 (1.2) |
| Admission days to each hospital from January to October 2020 (n, %) | Hospital A | 855 (5.2%) | 248 (4.8%) | 279 (5.2%) | 284 (5.6%) | 57,868 (15.3%) |
| | Hospital B | 2,846 (17.4%) | 959 (18.7%) | 1,121 (20.9%) | 1,076 (21.3%) | 37,358 (9.9%) |
| | Hospital C | 11,417 (69.6%) | 3,287 (64.1%) | 3,238 (60.3%) | 3,041 (60.2%) | 260,247 (68.7%) |
| | Hospital D | 1,279 (7.8%) | 634 (12.4%) | 731 (13.6%) | 653 (12.9%) | 23,226 (6.1%) |
| Number of SARS-CoV-2 PCR tests per patient (mean, SD) | | 2.7 (2.7) | 3.5 (3.2) | 3.8 (3.4) | 3.8 (3.4) | 0.9 (1.7) |
| Admission days to each ward type during infectious period[$] | General ward | 3,283 (87.5%) | 1,234 (96.7%) | 1,121 (96.9%) | 946 (96.5%) | — |
| | ICU/ HDU[*] | 471 (12.5%) | 42 (3.3%) | 36 (3.1%) | 34 (3.5%) | — |
| Admission days to each ward type during at-risk period[+] | General ward | 4,737 (96.4%) | 1,924 (95.1%) | 2,254 (94.7%) | 2,252 (94.9%) | 134,001 (91.8%) |
| | ICU/ HDU[*] | 178 (3.6%) | 100 (4.9%) | 125 (5.3%) | 122 (5.1%) | 11,968 (8.2%) |
| At-risk days per patient (mean, SD) | | 5.3 (11.5) | 15.2 (17.3) | 18.3 (18.1) | 19.8 (17.8) | 2.2 (10.3) |

[#]All patients included in the table had at least 1 day of inpatient stay during the observation period between 12 January and 2 October 2020.

[*]ICU/HDU, intensive care unit/high-dependency unit.

[$]Infectious period: Patients were considered infectious from the day after infection to 10 days after infection [21].

[+]At-risk period: Patients were considered to be at risk of being infected with SARS-CoV-2 from admission to either discharge/death or 4, 6, or 8 days before their first positive PCR tests, i.e., day of presumed infection.

Testing capacity increased substantially after the beginning of March 2020 (Fig 2A). The weekly incidence of newly detected SARS-CoV-2 infections in the 4 hospitals, including both community-acquired and nosocomial cases, peaked between March and May 2020.

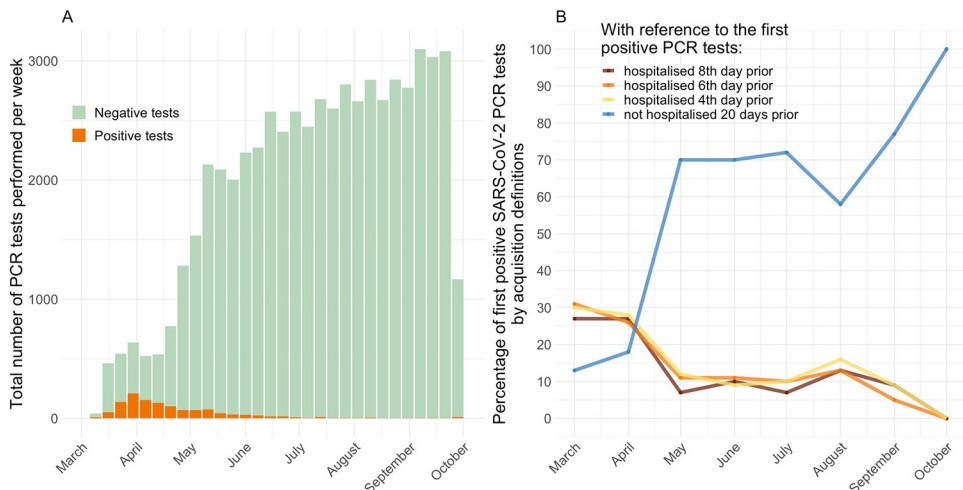

**Fig 2.** Weekly sums of SARS-CoV-2 PCR tests performed during March to October 2020 (Panel A). The stacked green bars indicate the number of negative tests. The stacked orange bars indicate the number of positive tests. Percentage of first positive SARS-CoV-2 PCR tests classified by different types of acquisition (Panel B). The colours represent patients who were inpatients on the eighth (red), sixth (orange), and fourth day (yellow) prior to their first positive tests, and who were not hospitalised in the 20 days prior to their first positive tests (blue). These classifications are not mutually exclusive, e.g., a patient who was admitted for 10 days continuously prior to the first positive PCR test would contribute to all first 3 groups. SARS-CoV-2, Severe Acute Respiratory Syndrome Coronavirus 2.

A total of 271 patients had at least 1 day of hospitalisation in the 20 days prior to being tested positive for SARS-CoV-2. Out of these patients, 130 (47.9%) were inpatients on their day of infection, based on an assumed incubation period of 5 days. Out of the 130 patients, 102 had at least 1 negative PCR test during day 1 to 5 of their hospitalisation (78.5%). The median length of stay for the admissions during which the patients were infected with SARS-CoV-2 was 21 days (interquartile range 13 to 35 days). The median day of hospitalisation when these patients were assumed to have been infected was day 8 (interquartile range 3 to 18 days).

## Healthcare worker characteristics

Out of a total of 13,514 HCWs in the 4 hospitals participating in HCW testing at some time, 5,596 worked on a single ward only such that their SARS-CoV-2 status could be considered with patients admitted to the same wards in the analysis (Table 2). During the study period, 11% (615/5,596) were positive for SARS-CoV-2. Among those who were positive, 57.4% (353/615) had a positive PCR test, while the rest were diagnosed based on serology.

The timelines of the numbers of susceptible patients and infectious patients and HCWs on each ward showed that most of the peaks in the number of potential transmission events occurred between March and June 2020 (S1 Fig). On most wards, there was a considerable overlap between the time series for infectious HCWs and patients and the time series of transmission events.

## Transmission risk to patients

We first used multivariable logistic regression to identify the factors associated with nosocomial transmission of SARS-CoV-2 to susceptible patients (Table 3). Infection risk reduced with the introduction of more stringent infection prevention and control measures in phase 3 (adjusted odds ratio (aOR) 0.25, 95% CI 0.14, 0.42) compared to phase 1. Presence of patients with hospital-acquired SARS-CoV-2 was associated with the highest risk of acquisition in

**Table 2. Characteristics of the HCWs included in the analysis.**

| | | Positive for SARS-CoV-2 $n$ = 615 | Negative for SARS-CoV-2 $n$ = 4,981 |
|---|---|---|---|
| **Age (mean age in years, SD)** | | 39.5 (11.1) | 39.6 (11.7) |
| **Sex (n, %)** | **Female** | 485 (78.9%) | 3,902 (78.3%) |
| | **Male** | 130 (21.1%) | 1,079 (21.7%) |
| **Roles (n, %)** | **Doctor** | 98 (15.9%) | 955 (19.2%) |
| | **Nurses** | 306 (49.8%) | 1,984 (39.8%) |
| | **Allied health** | 136 (22.1%) | 1,274 (25.6%) |
| | **Nonclinical staff** | 75 (12.2%) | 768 (15.4%) |
| **Hospital worked in during the observation period (n, %)** | **Hospital A** | 97 (15.8%) | 972 (19.5%) |
| | **Hospital B** | 91 (14.8%) | 454 (9.1%) |
| | **Hospital C** | 379 (61.6%) | 3,276 (65.8%) |
| | **Hospital D** | 48 (7.8%) | 279 (5.6%) |
| **Ward type worked in during the observation period (n, %)** | **General ward** | 569 (92.5%) | 4,384 (88.0%) |
| | **ICU/HDU**[*] | 46 (7.5%) | 597 (12.0%) |

[*]ICU/HDU, intensive care unit/high-dependency unit.

HCW, healthcare worker; SARS-CoV-2, Severe Acute Respiratory Syndrome Coronavirus 2.

susceptible patients (aOR, 1.76, 95% CI 1.51, 2.04), followed by the presence of infected HCWs (aOR 1.45, 95% CI 1.22, 1.71). The evidence that patients with community-onset COVID-19 were associated with increased transmission was weaker (aOR 1.12, 95% CI 0.96, 1.26).

To further investigate if the demographic variables and transmissions from infectious patients and HCWs adequately accounted for patient acquisition of SARS-CoV-2, we used these variables in a generalised additive model (Code block B in S3 Text). After adjusting for these variables, the results showed that the variation in the risk of nosocomial infection over the study period remained though at a reduced level, suggesting that transmission risks were incompletely accounted for (Fig 3A). We further used the above generalised additive model to explore how risk of nosocomial SARS-CoV-2 infection varied with day of hospitalisation (Fig A in S1 Text). This risk remained largely constant throughout a patient's hospital stay once the numbers of infectious patients and HCWs in the same ward were accounted for.

A feature of the logistic regression model is that it assumed the effect of each additional infectious patient or HCW as multiplicative. To improve interpretability and applicability of the estimates in a clinical setting, we used a statistical model that allows each extra infectious individual to increase the probability of infection in an additive way (a generalised mixed model with an identity link). Infectious patients on the same ward who were assumed to have hospital-acquired SARS-CoV-2 showed the strongest association with acquisition of nosocomial COVID-19 in susceptible patients (Fig 4). Using an assumed incubation period of 5 days, the absolute risk of acquiring SARS-CoV-2 per day of exposure to a patient with hospital-acquired SARS-CoV-2 infection was 0.75% (95% CrI 0.55% to 0.95%). The risks of acquiring SARS-CoV-2 per day of exposure to an infectious patient who acquired SARS-CoV-2 in the community or to an infectious HCW were smaller. One day of exposure to an infected HCW or patient with community-acquired COVID19 was associated with absolute risks of 0.20% (95% CrI 0.16% to 0.22%) and 0.17% (95% CrI 0.13% to 0.22%), respectively, for susceptible patients.

## Transmission risk to healthcare workers

We performed similar analyses to quantify the risk of transmission to HCWs. The multivariable logistic regression results showed that nurses were at the highest risk of being

**Table 3. Predictors of SARS-CoV-2 infection in admitted patients during their hospital stay from multivariable logistic regression results.**

| | Incubation period | | | | | | | | |
|---|---|---|---|---|---|---|---|---|---|
| | 5 days | | | 3 days | | | 7 days | | |
| Characteristics | aOR[1] | 95% CI[1] | p-value[1] | aOR[1] | 95% CI[1] | p-value[1] | aOR[1] | 95% CI[1] | p-value[1] |
| Age | 1.03 | 1.01, 1.04 | <0.001 | 1.02 | 1.01, 1.04 | <0.001 | 1.02 | 1.01, 1.04 | <0.001 |
| **Sex** | | | | | | | | | |
| **Female** | 1[2] | — | | 1 | — | | 1 | — | |
| **Male** | 1.03 | 0.69, 1.52 | 0.90 | 0.97 | 0.64, 1.44 | 0.90 | 1.02 | 0.68, 1.54 | 0.91 |
| **Ethnic group** | | | | | | | | | |
| **White** | 1 | — | | 1 | — | | 1 | — | |
| **Non-white** | 0.49 | 0.08, 1.61 | 0.30 | 0.00 | 0.00, 0.29 | 0.97 | 0.52 | 0.08, 1.71 | 0.40 |
| **Infectious patients with community-acquired SARS-CoV-2 on the same ward** | 1.12 | 0.96, 1.26 | 0.10 | 1.04 | 0.89, 1.18 | 0.60 | 1.27 | 1.08, 1.44 | <0.001 |
| **Infectious patients with hospital-acquired SARS-CoV-2 on the same ward** | 1.76 | 1.51, 2.04 | <0.001 | 1.94 | 1.64, 2.28 | <0.001 | 1.62 | 1.35, 1.91 | <0.001 |
| **Infectious staff on the same ward** | 1.45 | 1.22, 1.71 | <0.001 | 1.55 | 1.21, 1.94 | <0.001 | 1.46 | 1.27, 1.67 | <0.001 |
| **Hospital** | | | | | | | | | |
| **Hospital A** | 1 | — | | 1 | — | | 1 | — | |
| **Hospital B** | 2.06 | 0.87, 5.70 | 0.13 | 2.58 | 1.02, 7.87 | 0.06 | 3.22 | 1.22, 11.1 | 0.03 |
| **Hospital C** | 1.65 | 0.75, 4.33 | 0.30 | 2.01 | 0.87, 5.85 | 0.14 | 2.37 | 0.95, 7.92 | 0.10 |
| **Hospital D** | 3.06 | 1.26, 8.58 | 0.02 | 3.94 | 1.54, 12.1 | 0.01 | 3.96 | 1.39, 14.1 | 0.01 |
| **Type of ward** | | | | | | | | | |
| **General ward** | 1 | — | | 1 | — | | 1 | — | |
| **ICU/HDU[3]** | 0.62 | 0.15, 1.68 | 0.40 | 0.45 | 0.07, 1.44 | 0.30 | 0.20 | 0.01, 0.90 | 0.11 |
| **Day of stay** | 1.00 | 0.99, 1.01 | 0.70 | 1.00 | 0.99, 1.01 | 0.98 | 0.99 | 0.98, 1.00 | 0.10 |
| **Phases[4]** | | | | | | | | | |
| **1** | 1 | — | | 1 | — | | 1 | — | |
| **2** | 2.06 | 1.15, 3.62 | 0.013 | 2.80 | 1.55, 4.98 | 0.001 | 1.30 | 0.69, 2.38 | 0.40 |
| **3** | 0.25 | 0.14, 0.42 | <0.001 | 0.27 | 0.16, 0.46 | 0.002 | 0.30 | 0.18, 0.50 | <0.001 |

All independent variables used in the multivariable regression model are listed in the table. The corresponding univariable analysis is presented in Table A in S1 Text.

[1]aOR, adjusted odds ratio; CI, confidence interval. p-Values were calculated with the Wald test.

[2]Value of 1 for aOR represents the comparison group for categorical variables.

[3]ICU/HDU, intensive care unit/high-dependency unit.

[4]In addition to phases, calendar days was included as a nonlinear independent variable in the logistic regression, fitted with a linear spline function with 2 knots.

infected with SARS-CoV-2 (aOR 1.54, 95% CI 1.17, 2.04) compared with doctors. Working in the intensive care or high-dependency units was protective against transmission (aOR 0.55, 95% CI 0.39, 0.76) compared to the general ward. Transmission risk reduced in phases 2 and 3 (aOR 0.32, 95% CI 0.21, 0.37, and 0.63 and 95% CI 0.49, 0.81, respectively) compared to phase 1. The number of infectious HCWs and patients who had

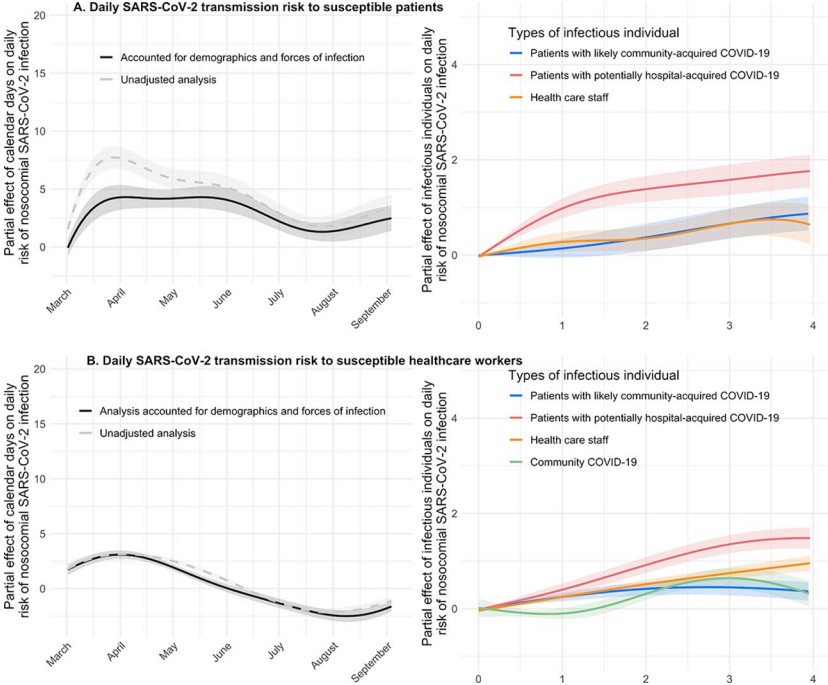

**Fig 3.** Daily transmission risk to susceptible patients (Panel A) and HCWs (Panel B) using a generalised additive model with a logit link. The smooth, nonlinear partial effects of calendar day, infectious patients, and HCWs on the daily risk of nosocomial SARS-CoV-2 infection are presented as coloured lines. These partial effects are the isolated effects of each group of infectious individuals on the binary outcome of assumed acquisition (yes/no) on each day as the dependent variable. Infectious patients and HCWs were both associated with increased risk of nosocomial infection. The presence of more infectious patients or HCWs in a ward on a given day was associated with higher transmission risk. COVID-19, Coronavirus Disease 2019; HCW, healthcare worker; SARS-CoV-2, Severe Acute Respiratory Syndrome Coronavirus 2.

hospital-acquired SARS-CoV-2 on the same ward had the strongest associations with transmission to HCWs (aOR 1.33, 95% CI 1.21, 1.45 and aOR 1.45, 95% CI 1.34, 1.55, respectively) (Table 4).

Using the additive statistical model (Fig 5), the strongest association was with other infectious staff and patients with hospital-acquired SARSCoV-2. However, the additional risks posed by exposures to these infectious HCWs and patients to the susceptible HCWs were less compared to that for susceptible patients. A single day of exposure to infected HCWs and patients with hospital-acquired SARS-CoV-2 patients on the same ward was associated with an increased absolute daily risk of 0.08% (95% CrI 0.03% to 0.16% and 0.06% to 0.10%, respectively). The corresponding increase in absolute daily risk from a day of exposure to an infected patient with community-acquired SARS-CoV-2 was 0.02% (95% CrI 0.02% to 0.02%).

The background transmission risks to HCWs including that from community sources and undetected cases among both HCWS and patients were slightly less than those observed in the patients. The contribution of these undetected cases to the daily risk of SARS-CoV-2 acquisition was about 0.03% (95% CrI 0.02% to 0.03%) and 0.01% (95% CrI 0.01% to 0.01%) for the susceptible patients and HCWs, respectively. Findings from sensitivity analyses, which excluded data from phase 3 and using different prior distributions, gave similar results as the main analyses (Tables A and B in S2 Text).

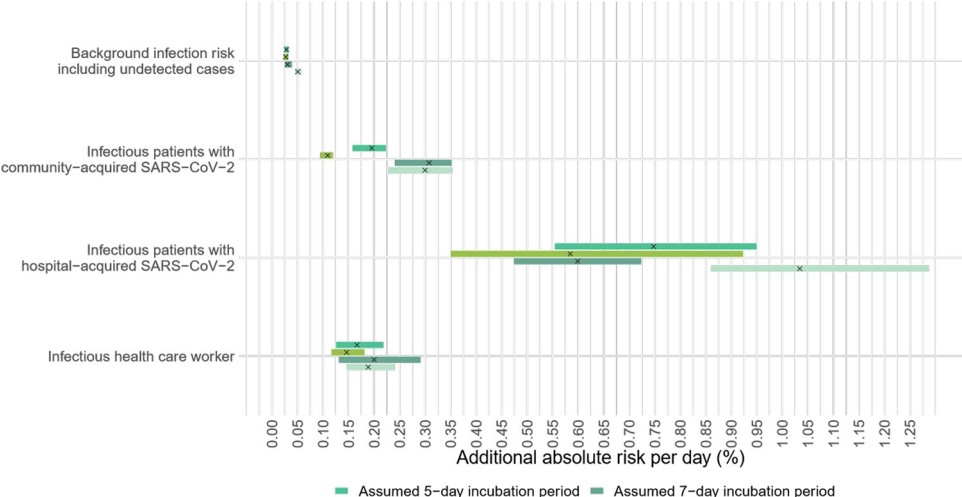

**Fig 4.** Additional risk of suspected nosocomial acquisition of SARS-CoV-2 experienced by a single susceptible patient contributed by (i) infectious patients who acquired SARS-CoV-2 in the community (second row); (ii) infectious patients who acquired SARS-CoV-2 in the hospital (third row); and (iii) infectious HCWs (last row). A generalised mixed model with an identity link was used, with assumed nosocomial acquisition (yes/no) on each day as the dependent variable. Both the intercepts and slopes were allowed to vary by ward. The top row shows the variation of the intercepts of the model, which represent the background infection risk posed by infectious patients and HCWs who are undetected. Each horizontal bar represents the 95% CrI of the estimate. The black crosses in the centre of each bar represent the median of the estimates. CrI, credible interval; HCW, healthcare worker; SARS-CoV-2, Severe Acute Respiratory Syndrome Coronavirus 2.

## Discussion

The consistent finding in the above analysis is that the patients who acquired SARS-CoV-2 in the hospital and, to a lesser degree, infectious HCWs likely working prior to the onset of symptoms were the most strongly associated with increased risk of SARS-CoV-2 transmission in the hospital setting. In contrast, exposure to patients who had acquired SARS-CoV-2 in the community appeared to be associated with more modest increases in the daily risk of infection for both healthcare staff and the other patients. We found evidence of a dose–response effect: Exposure to more infectious patients and healthcare staff were both associated with increasing daily risk of acquiring SARS-CoV-2.

These findings can parsimoniously be explained by newly infected individuals having high transmission potential to patients and staff. Multiple lines of evidence indicate that a substantial proportion of transmission precedes symptom onset and point to rapidly declining infectiousness with time since symptom onset [21,28]. Secondly, patients who acquired SARS-CoV-2 in the community are more likely to first present with symptoms compatible with COVID-19 upon admission and be rapidly segregated from the susceptible population with careful implementation of infection prevention and control guidelines.

To our knowledge, our study is the first analysis of a large dataset consisting of both hospitalised patients and HCWs at individual level for the quantification of transmission pathways of SARS-CoV-2 in the hospital setting. We searched the PubMed database using the search terms ("COVID-19" OR "SARS-CoV-2") AND ("nosocomial" OR "hospital") AND ("transmission") in either the abstracts or titles, for English-language articles published up to 31 March 2021. This returned 748 results, out of which 10 reported transmission events in the hospital setting quantitatively. These publications can be broadly categorised to epidemiological descriptions of isolated outbreaks (5) or contact tracing of patients exposed to infected

**Table 4. Predictors of SARS-CoV-2 infection in HCWs from multivariable logistic regression.**

| | Incubation period | | | | | | | | |
| --- | --- | --- | --- | --- | --- | --- | --- | --- | --- |
| | 5 days | | | 3 days | | | 7 days | | |
| Characteristics | aOR[1] | 95% CI[1] | p-value[1] | aOR[1] | 95% CI[1] | p-value[1] | aOR[1] | 95% CI[1] | p-value[1] |
| Age[2] | 1.00 | 0.99, 1.01 | 0.9 | 1.00 | 0.99, 1.01 | 0.9 | 1.00 | 0.99, 1.01 | 0.92 |
| **Sex** | | | | | | | | | |
| Female | 1[3] | — | | 1 | — | | 1 | — | |
| Male | 1.23 | 0.97, 1.55 | 0.09 | 1.19 | 0.93, 1.50 | 0.2 | 1.18 | 0.93, 1.49 | 0.2 |
| **Role** | | | | | | | | | |
| Doctor | 1 | — | | 1 | — | | 1 | — | |
| Nurse | 1.54 | 1.17, 2.04 | 0.002 | 1.63 | 1.24, 2.18 | <0.001 | 1.48 | 1.13, 1.95 | 0.005 |
| Allied health | 1.02 | 0.75, 1.39 | 0.9 | 1.05 | 0.77, 1.44 | 0.7 | 0.92 | 0.68, 1.24 | 0.6 |
| Nonclinical staff | 1.02 | 0.71, 1.44 | 0.9 | 1.08 | 0.76, 1.54 | 0.7 | 0.93 | 0.65, 1.31 | 0.7 |
| **Infectious cases in the community** | 1.28 | 1.23, 1.33 | <0.001 | 1.36 | 1.30, 1.42 | <0.001 | 1.23 | 1.18, 1.28 | <0.001 |
| **Infectious patients with community-acquired SARS-CoV-2 on the same ward** | 0.99 | 0.92, 1.05 | 0.7 | 1.01 | 0.96, 1.05 | 0.8 | 0.95 | 0.85, 1.04 | 0.3 |
| **Infectious patients with hospital-acquired SARS-CoV-2 on the same ward** | 1.33 | 1.21, 1.45 | <0.001 | 1.46 | 1.33, 1.59 | <0.001 | 1.32 | 1.20, 1.44 | <0.001 |
| **Infectious staff on the same ward** | 1.45 | 1.34, 1.55 | <0.001 | 1.49 | 1.34, 1.66 | <0.001 | 1.40 | 1.32, 1.48 | <0.001 |
| **Hospital** | | | | | | | | | |
| Hospital A | 1 | — | | 1 | — | | 1 | — | |
| Hospital B | 1.65 | 1.19, 2.28 | 0.002 | 1.74 | 1.26, 2.38 | <0.001 | 1.66 | 1.20, 2.28 | 0.002 |
| Hospital C | 1.22 | 0.96, 1.56 | 0.11 | 1.13 | 0.89, 1.46 | 0.3 | 1.20 | 0.95, 1.54 | 0.13 |
| Hospital D | 1.40 | 0.92, 2.09 | 0.10 | 1.46 | 0.97, 2.16 | 0.061 | 1.30 | 0.85, 1.95 | 0.2 |
| **Type of ward** | | | | | | | | | |
| General ward | 1 | — | | 1 | — | | 1 | — | |
| ICU/HDU[4] | 0.55 | 0.39, 0.76 | <0.001 | 0.57 | 0.41, 0.79 | <0.001 | 0.55 | 0.39, 0.75 | <0.001 |
| **Phase[5]** | | | | | | | | | |
| 1 | 1 | — | | 1 | 1 | | 1 | — | |
| 2 | 0.32 | 0.21, 0.47 | <0.001 | 0.22 | 0.14, 0.33 | <0.001 | 0.32 | 0.22, 0.46 | <0.001 |
| 3 | 0.63 | 0.49, 0.81 | <0.001 | 0.74 | 0.57, 0.96 | <0.001 | 0.53 | 0.42, 0.67 | <0.001 |

All independent variables used in the multivariable regression model are listed in the table. The corresponding univariable analysis is presented in Table B in S1 Text.

[1] aOR, adjusted odds ratio; CI, confidence interval. p-Values were calculated with the Wald test.

[2] Age measured in years.

[3] Value of 1 for aOR represents the comparison group for categorical variables.

[4] ICU/HDU, intensive care unit/high-dependency unit.

[5] In addition to phases, calendar days was included as a nonlinear independent variable in the logistic regression, fitted with a linear spline function with 2 knots.

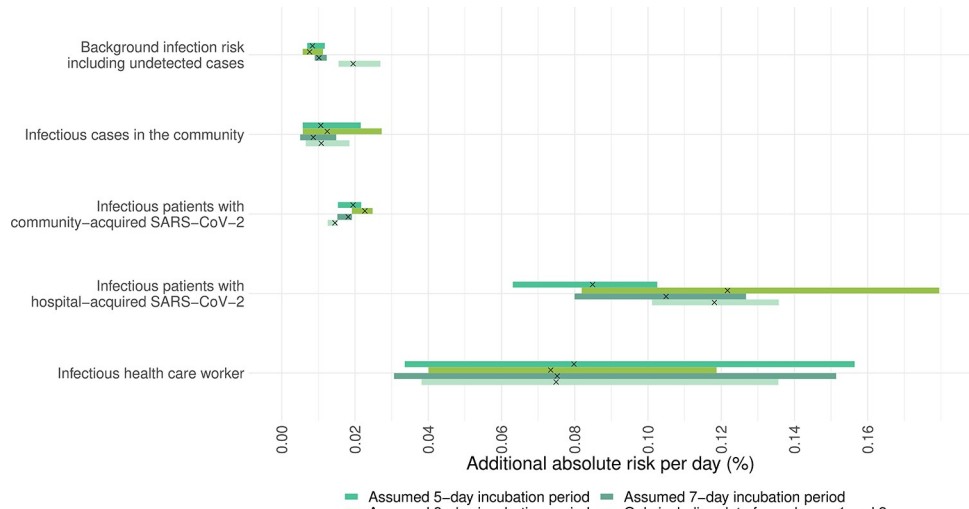

**Fig 5.** Additional risk of suspected nosocomial acquisition of SARS-CoV-2 experienced by a single susceptible HCW contributed by (i) infectious patients who acquired SARS-CoV-2 in the community (second row); (ii) infectious patients who acquired SARS-CoV-2 in the hospital (third row); and (iii) infectious HCWs (last row). A generalised mixed model with an identity link was used, with assumed nosocomial acquisition (yes/no) on each day as the dependent variable. Both the intercepts and slopes were allowed to vary by ward. The top row shows the variation of the intercepts of the model, which represent the background infection risk posed by infectious patients and HCWs who are undetected. Each horizontal bar represents the 95% CrI of the estimate. The black crosses in the centre of each bar represent the median of the estimates. CrI, credible interval; HCW, healthcare worker; SARS-CoV-2, Severe Acute Respiratory Syndrome Coronavirus 2.

HCWs (1), retrospective cohort studies involving a particular group of patients, e.g., patients who underwent operations (2), and using genomic sequencing to identify transmission clusters (2). None of the studies reported the comparative transmission rates among patients and staff.

There are several limitations in our analysis. Firstly, the dates on which the infected patients and HCWs first developed symptoms were not available. Hence, we needed to assume that the PCR test swabs were taken on the symptom onset dates. While this assumption is reasonable based on the analysis of a subset of data early in the pandemic, it is not true from phase 3 onwards when weekly screening of patients regardless of symptoms was implemented. In HCW infections identified with serology, self-reported symptom onset dates may potentially suffer from recall bias. We addressed this by performing sensitivity analysis comparing model outputs when using data collected during phases 1 and 2 versus phase 3 (Tables A and B in S2 Text). Secondly, we assumed that HCWs were absent from work after the dates on which their first positive PCR test swabs were taken or COVID-19 symptoms were first self-reported. However, where HCWs experienced minimal or no symptoms they may have continued to work. These issues could be further explored using HCW absentee data in subsequent analysis. Lastly, we did not consider the genomic sequences of the SARS-CoV-2 viruses to confirm the transmission pathways. A proportion of these infections in the HCWs could potentially be introduced from the community. We attempted to address this by using admission rate of community-acquired cases to extrapolate community infection pressure, which overcomes the issue of low community testing in the initial phase of the pandemic.

A key challenge in this analysis is that the times of infection are unknown. This has led to the adoption of various arbitrary cutoffs on length of stay prior to infection to define nosocomial infection. Further analysis using data augmentation methods, incorporating the PCR

cycle threshold values, may potentially overcome this to produce estimates that better account for different sources of uncertainty. Other drivers of SARS-CoV-2 transmissions in the hospital setting not fully explained by infection pressures, which we did not capture in the analysis, may include variation in ward occupancy, community-acquired cases who did not develop symptoms until after hospitalisation, change in nature or frequency of SARS-CoV-2 exposures throughout hospitalisation, or could reflect frailties, i.e., those patients who have stayed 20 days and not been infected may be at lower risk of infection. However, recent work using detailed epidemiological and genomic data to infer transmission networks echoed our main finding that patients are more likely to be infected by other patients than by HCWs [29,30].

Our findings support enhanced infection prevention and control efforts to prevent and identify early hospital-onset SARS-CoV-2 infection. Where either community or local ward prevalence is sufficiently high and resources permit, regular screening and prompt testing and identification of such patients are important. Similarly, measures to ensure symptomatic staff are not at work, including adequate sick pay arrangements, are vital. Regular staff screening is also likely to reduce transmission. Staff acquisition and transient asymptomatic carriage, contamination of equipment, and the general environment or the air are possible mediators of transmission events assigned in the analysis as patient-to-patient and need further investigation. The relatively low risk of transmission associated with patients with suspected community-acquired COVID-19 suggests that for these patients, the peak of their infectivity may have passed such that existing infection prevention and control policies including universal PPE, prompt testing, and isolation of suspected or known cases [16] are sufficient to mitigate most of the remaining infectiousness. Our analysis shows that despite these measures, patients and staff are at risk from newly infected individuals. Due to the difficulties in disentangling the effect of infection prevention and control measures and surges in SARS-CoV-2 in the community setting, we cannot provide conclusive evidence on how interventions around hospital-onset cases could be enhanced. However, others have suggested that enhanced PPE for HCWs and ventilation may play a role [4,31–33].

In conclusion, our data provide strong evidence that newly infected patients are associated with a high risk of onward transmission to patients and HCWs in hospital. Further investigation is needed into how best to enhance infection control and prevention efforts around these patients.

## Transparency statement

The corresponding author affirms that the manuscript is an honest, accurate, and transparent account of the study being reported; that no important aspects of the study have been omitted; and that any discrepancies from the study as originally planned (and, if relevant, registered) have been explained.

## Contributor and guarantor information

DWE and BSC conceptualized this work. YM, DWE and BSC performed the statistical analysis. YM drafted the first version of the manuscript. DWE, YM and KJ verified the underlying data. All authors reviewed and edited subsequent versions of the manuscript. The corresponding author attests that all listed authors meet authorship criteria and that no others meeting the criteria have been omitted. The corresponding author accepts full responsibility for the work and/or the conduct of the study, had access to the data, and controlled the decision to publish.

## Patient and public involvement

Patients and the public were not involved in the design, analysis, and reporting of this study. The study results will be disseminated to the public communities through publication.

## Copyright/licence for publication

The Corresponding Author has the right to grant on behalf of all authors and does grant on behalf of all authors, a worldwide licence to the Publishers and its licensees in perpetuity, in all forms, formats and media (whether known now or created in the future), to (i) publish, reproduce, distribute, display, and store the Contribution; (ii) translate the Contribution into other languages, create adaptations, reprints, include within collections, and create summaries, extracts, and/or abstracts of the Contribution; (iii) create any other derivative work(s) based on the Contribution; (iv) to exploit all subsidiary rights in the Contribution; (v) the inclusion of electronic links from the Contribution to third party material wherever it may be located; and (vi) licence any third party to do any or all of the above.

For the purpose of Open Access, the author has applied a CC BY public copyright licence to any Author Accepted Manuscript version arising from this submission.

## Supporting information

**S1 Checklist. STROBE checklist.**
(PDF)

**S1 Fig. The timelines of potential nosocomial transmission events and the numbers of infectious patients and HCWs.** HCW, healthcare worker; SARS-CoV-2, Severe Acute Respiratory Syndrome Coronavirus 2.
(DOCX)

**S2 Fig. Distributions of incubation period and generation time used in the analysis.**
(DOCX)

**S3 Fig. Infection pressure from the community was extrapolated from the number of patients with community-acquired SARS-CoV-2 infection admitted to the hospitals.** COVID-19, Coronavirus Disease 2019; SARS-CoV-2, Severe Acute Respiratory Syndrome Coronavirus 2.
(DOCX)

**S1 Text. Logistic regression (Model 1) results, generalised additive model (Model 2) results, and generalised linear model with identity link (Model 3) model assessment and comparisons.**
(DOCX)

**S2 Text. Generalised linear model with identity link (Model 3) model sensitivity analysis.**
(DOCX)

**S3 Text. Model R codes.**
(DOCX)

**S4 Text. Details of real-time polymerase chain reaction (RT-PCR) performed with combined nasal and oropharyngeal swabs.**
(DOCX)

**S5 Text. Members of the Oxford COVID infection review team.**
(DOCX)

## Acknowledgments

The authors acknowledge valuable contributions from Omar Risk, Hannah Chase, Ishta Sharma, Sarah Peters, Tamsin Cargill, Grace Barnes, Josh Hamblin, Jenny Tempest-Mitchell, Archie Lodge, Sai Arathi Parepalli, Raghav Sudarshan, Hannah Callaghan, Imogen Vorley, Ashley Elder, Danica Fernandes, Gurleen Kaur, Bara'a Elhag, Edward David, Rumbi Mutenga, Dylan Riley, Emel Yildirim, Maria Tsakok, and Naomi Hudson from Oxford University Hospitals NHS Foundation Trust and University of Oxford Medical School for data collection.

The views expressed in this publication are those of the authors and not necessarily those of the UK National Health Service, the National Institute for Health Research, the Department of Health or Public Health England, and other funders. All authors confirm that we had full access to all the data in the study and accept responsibility to submit for publication.

## Author Contributions

**Conceptualization:** Yin Mo, David W. Eyre, Ben S. Cooper.

**Data curation:** David W. Eyre.

**Formal analysis:** Yin Mo, David W. Eyre, Ben S. Cooper.

**Funding acquisition:** David W. Eyre, Timothy M. Walker, Katie Jeffery.

**Investigation:** Yin Mo, David W. Eyre, Sheila F. Lumley, Timothy M. Walker, Robert H. Shaw, Denise O'Donnell, Lisa Butcher, Katie Jeffery, Ben S. Cooper.

**Methodology:** Yin Mo, David W. Eyre, Christl A. Donnelly, Ben S. Cooper.

**Project administration:** David W. Eyre, Sheila F. Lumley, Timothy M. Walker, Katie Jeffery.

**Supervision:** David W. Eyre, Ben S. Cooper.

**Validation:** Yin Mo, David W. Eyre, Ben S. Cooper.

**Visualization:** Yin Mo, David W. Eyre, Ben S. Cooper.

**Writing – original draft:** Yin Mo.

**Writing – review & editing:** Yin Mo, David W. Eyre, Sheila F. Lumley, Timothy M. Walker, Robert H. Shaw, Denise O'Donnell, Lisa Butcher, Katie Jeffery, Christl A. Donnelly, Ben S. Cooper.

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
