## [Editor Report · Decision Letter 0]

11 May 2021

Dear Dr Yin, 

Thank you for submitting your manuscript entitled "Identifying the drivers of SARS-CoV-2 transmission in the hospital setting: an observational cohort study" for consideration by PLOS Medicine.

Your manuscript has now been evaluated by the PLOS Medicine editorial staff and I am writing to let you know that we would like to send your submission out for external peer review.

Please re-submit your manuscript within two working days, i.e. by May 13 2021 11:59PM.

Kind regards,

Louise Gaynor-Brook, MBBS PhD

Senior Editor

PLOS Medicine

---

## [Decision Letter · Decision Letter 1]

4 Jul 2021

Dear Dr. Yin,

Thank you very much for submitting your manuscript "Identifying the drivers of SARS-CoV-2 transmission in the hospital setting: an observational cohort study" (PMEDICINE-D-21-02091R1) for consideration at PLOS Medicine. 

Your paper was sent to four independent reviewers, including a statistical reviewer. It has been discussed among all the editors here and with an academic editor with relevant expertise. The reviews are appended at the bottom of this email and any accompanying reviewer attachments can be seen via the link below:

[LINK]

In light of these reviews, I am afraid that we will not be able to accept the manuscript for publication in the journal in its current form, but we would like to consider a revised version that addresses the reviewers' and editors' comments. Obviously we cannot make any decision about publication until we have seen the revised manuscript and your response, and we plan to seek re-review by one or more of the reviewers. 

We expect to receive your revised manuscript by Jul 23 2021 11:59PM. Please email us (plosmedicine@plos.org) if you have any questions or concerns.

We look forward to receiving your revised manuscript. 

Sincerely,

Louise Gaynor-Brook, MBBS PhD

Associate Editor 

PLOS Medicine

plosmedicine.org

Comments from the Academic Editor:

I think the study is interesting and relevant. What is maybe most problematic is the assumption that infected HCW were infected in the hospital. I believe that most transmission in that time period took place in the community, and I would guess that most HCWs got infected in the community. I agree with the reviewers that there needs to be more information on how the setup is of the hospitals from which the data came, and what the implemented infection prevention strategies were. Also in particular, how these hospitals dealt with COVID patients and with HCWs showing any symptoms. 

General comments:

Throughout the paper, please adapt reference call-outs to the following style: "... since symptom onset [21,26]." (noting the absence of spaces within the square brackets).

Title: Please revise your title according to PLOS Medicine's style. Please place the study design in the subtitle (ie, after a colon). We suggest “Transmission of community- and hospital-acquired SARS-CoV-2 in the hospital setting: a cohort study” or similar 

Abstract Background: Provide expand upon the context of why the study is important. The final sentence should clearly state the study question.

Abstract Methods and Findings:

Please specify the study design.

Please provide brief demographic details of the study population (e.g. sex, age, ethnicity, etc)

Please define CrI at first use.

Should eight infections per 1000 susceptible patients also be per day (as for 2/1000 susceptible patients/day, mentioned later)?

In the last sentence of the Abstract Methods and Findings section, please describe 2-3 of the main limitation(s) of the study's methodology.

Abstract Conclusions:

Please begin your Abstract Conclusions with "In this study, we observed ..." or similar, to summarize the main findings from your study, emphasizing what is new without overstating your conclusions.

Author Summary:

In the final bullet point of ‘What Do These Findings Mean?’, please describe the main limitation(s) of the study in non-technical language.

Methods:

Did your study have a prospective protocol or analysis plan? Please state this (either way) early in the Methods section. If a prospective analysis plan (from your funding proposal, IRB or other ethics committee submission, study protocol, or other planning document written before analyzing the data) was used in designing the study, please include the relevant prospectively written document with your revised manuscript as a Supporting Information file to be published alongside your study, and cite it in the Methods section. A legend for this file should be included at the end of your manuscript. If no such document exists, please make sure that the Methods section transparently describes when analyses were planned, and if/when reported analyses differed from those that were planned. Changes in the analysis-- including those made in response to peer review comments-- should be identified as such in the Methods section of the paper, with rationale. If a reported analysis was performed based on an interesting but unanticipated pattern in the data, please be clear that the analysis was data-driven.

Please provide the name(s) of the institutional review board(s) that provided ethical approval for your study.

Thank you for providing a completed STROBE checklist in the Supporting Information. Please add the following statement, or similar, to the Methods: "This study is reported as per the Strengthening the Reporting of Observational Studies in Epidemiology (STROBE) guideline (S1 Checklist)." Please revise your checklist to use section and paragraph numbers, rather than page numbers.

Line 188 - please clarify whether “level-2” PPE continued to be used for aerosol generating procedures or in other circumstances after 1 April 2020 

Results: 

Line 281 (paragraph ‘Transmission risk to patients’) - For the adjusted analyses, please also provide the unadjusted analyses and indicate the factors which have been adjusted for. 

The OR results presented in Table 3 & 4 are presented in the main text as aOR - please clarify. 

Line 324 (paragraph ‘Transmission risk to healthcare workers’) - For the adjusted analyses, please also provide the unadjusted analyses and indicate the factors which have been adjusted for. 

Please clarify what the different phases (1-3) are in the main text of your results.

Where OR / aOR are presented, please specify the comparison group.

Discussion:

Please present and organize the Discussion as follows: a short, clear summary of the article's findings; what the study adds to existing research and where and why the results may differ from previous research; strengths and limitations of the study; implications and next steps for research, clinical practice, and/or public policy; one-paragraph conclusion.

Figures:

Please consider avoiding the use of red and green in order to make your figure more accessible to those with colour blindness.

Tables:

Tables 3 & 4 - please provide both OR and aOR results. The OR results presented in Tables 3 & 4 are presented in the main text as aOR - please clarify. Please specify the variables controlled for in the table legends. 

When a p value is given, please specify the statistical test used to determine it in the respective figure legend

Supplementary files: 

Please provide titles and legends for each individual table and figure in the Supporting Information.

Tables S1 & S2 - please provide both OR and aOR results. Please specify the variables controlled for in the table legends. When a p value is given, please specify the statistical test used to determine it in the respective figure legend

Please rename your supplementary files according to PLOS Medicine's style - Figure S1, Table S1 (rather than using subsections) and please ensure that all supplementary files are referred to in the main text of your manuscript e.g. Table S1 is not referred to

Comments from the reviewers:

Reviewer #1: Thank you for inviting me to review this manuscript. Mo and colleagues analyzed patient and healthcare worker (HCW) data from 4 UK hospitals during their first COVID-19 surge (Jan-Oct 2020) in order to identify associations between various clinical factors and hospital-onset SARS-CoV-2 infections in patients and HCWs. The authors used a series of statistical models to identify the following primary findings: 1) for hospitalized patients, spending time in the same ward as another patient with hospital-acquired SARS-CoV-2 was associated with a much higher risk of infection compared to being on the same ward as a patient with community-acquired SARS-CoV-2 or being exposed to an infectious HCW; 2) for HCWs, exposures to an infectious patient with hospital-acquired SARS-CoV-2 or another infectious HCW were associated with higher infection risk than exposure to a patient with community-acquired infection. Overall, this paper has a wealth of data and addresses an important topic in trying to better understand the factors that drive nosocomial infection. However, I have some concerns about their methods and assumptions.

Major Comments:

1. An underlying assumption seems to be that most or all HCW SARS-CoV-2 infections were acquired in the hospital, either from patients or other HCWs. However, it is quite clear now that many, if not the majority, of HCW infections are acquired in the community rather than in the hospital. Put another way, I would expect the number of infected patients and HCWs on the same ward to be highly correlated with community incidence; at the same, high community incidence will translate into a higher risk of HCWs acquiring infection outside of the hospital. Unless I am missing something, I do not see any attempt to account for household or non-hospital exposures and to adjust for community incidence rates.

2. A major weakness is that no genetic sequencing was done to confirm any of the presumed transmission pathways. As above, it is entirely plausible if not likely that many HCWs were acquired outside the hospital. At the least, this needs to be acknowledged, and the language around in-hospital "transmissions" needs to be substantially softened.

3. Please describe the room set-up in the hospitals, as this is highly relevant when making the claim that patients infected other patients. In particular, are all rooms single hospital rooms or are they shared rooms? Were patients with suspected or confirmed COVID-19 placed in airborne infection isolation rooms or simply into standard rooms? Do patients congregate in shared areas? If patients are not in shared rooms, I think a more likely explanation for the association between hospital-onset infections with new infections is spread via intermediary HCWs. 

4. Were PCR cycle threshold values available at all, and if so was there any association with the risk of transmission? High viral loads have been shown to be highly correlated with transmission risk in other studies. 

5. Was there any consideration that positive PCR tests with high Ct values in later months of the study (when much of the population would have been infected) might represent residual RNA from prior infections rather than active, contagious infections? Recent data suggest that a large fraction of positive PCR tests in hospitalized patients after the first surge represent residual RNA from prior infections. 

6. I'm a bit concerned about the fact that 43% of HCW infections were identified based on serology and memory of their symptoms. How reliable is this? 

7. The testing criteria for HCWs appeared to have been new symptoms, as well elective asymptomatic q2 week testing starting in late April 2020. Was there any requirement or recommendation for staff to get tested after unprotected exposure to patients or staff with COVID-19? 

Reviewer #2: See attachment

Michael Dewey

Reviewer #3: Dr. Mo and colleagues present very interesting data quantifying the frequency of potential hospital-acquired SARS-CoV-2 and quantifying the differential risk of infection via potential exposure to community-acquired vs hospital-acquired vs healthcare worker cases. These data are valuable because while nosocomial spread of SARS-CoV-2 has been well reported it has not been well characterized in terms of incidence and risk factors. Strengths of the study include drawing data from multiple hospitals, detailed daily level analyses, and differentiation of different kinds of exposures. Weaknesses include the indirectness of attribution of infections - these are imputed from associations rather than confirmed through detailed case-by-case analyses or sequencing. Nonetheless, the results are informative and welcome.

1. Text (line 234) says 97/920 patients tested positive for SARS-CoV-2 on day 7 or later but Table 1 says 120/920 patients tested positive assuming incubation period of 7 days. Please clarify this discrepancy. Does the latter include patients who tested positive post-discharge? If so, shouldn't the denominator be larger?

2. Do the authors have access to cycle thresholds? Comparing these values across community onset, hospital-acquired, and HCW infections could support their evaluations of infectiousness of these different groups

3. Can the authors say more about their baseline infection prevention infrastructure. Are patients in shared rooms or private rooms? What are the ventilation standards for hospital rooms? Are patients required to wear masks in the hospital? Were visitors permitted and if so were they tested?

4. Did the authors evaluate differences in risk in patients who shared a room with someone with hospital-acquired SARS-CoV-2 versus those who were simply on the same ward?

5. Were positive healthcare workers and patients interviewed to try to determine their particular exposures? Any patterns evident in interview-based exposure analyses?

Reviewer #4: Mo Y et al. report their results regarding their investigations into transmission pathways of SARS-CoV-2 among hospitalized patients and healthcare workers at four teaching hospitals in Oxfordshire. They found exposure to patients with hospital-acquired-SARS-CoV-2 to pose a substantial risk of subsequent infection leading to the conclusion that infection control measures to protect both patients and healthcare workers need to be optimized.

The topic is important and of interest to the readers of PLOS Medicine. 

Yet, I have the following concerns:

* Among healthcare workers testing positive for SARS-CoV-2, 43% of the diagnosis were based on serology - how was the time of infection determined in these cases (given the intrinsic limitations of drawing conclusions regarding time of infection based on serology)

* One of the main conclusions is that presence of patients with hospital-acquired SARS-CoV-2 was associated with the highest risk of acquisition in susceptible patients - given the nature of the data it is unclear how cause and effect can be distinguished here - how did the authors distinguish between all patients on a specific ward having been infected by one common index from ongoing exposure to subsequently infected patients? The current statement suggests that presence of patients with healthcare-associated SARS-CoV-2 infection puts other patients at risk for subsequent infection, yet all patients on a specific ward may have been infected by the same common source (resulting in differing times to detection, based on different timing of exposure and incubation periods).

* Presumably, patients with community-onset SARS-CoV-2 infection, were identified early during their course of hospitalization and placed under isolation precautions - it therefore seems clear that the risk of subsequent transmission should be reduced. The analyses does not seem to account for this and suggests an assumption of an equal risk which does not seem justifiable.

* How was cause and effect distinguished when analyzing transmission risks to healthcare workers? It is extremely challenging to analyze direction of transmission (i.e. from infected healthcare worker to patient or vice versa) - the data suggests that direction of transmission was clear in this dataset - how was it determined?

* It is unclear, why the dates on symptom onset for patients were not available - it is comment practice to document such information in medical records. As this data would enhance the quality of the study substantially, the authors should try to collect this information or explain in more detail why it is missing.

[LINK]

---

## [Decision Letter · Decision Letter 2]

3 Sep 2021

Dear Dr. Yin,

Thank you very much for re-submitting your manuscript "Transmission of community- and hospital-acquired SARS-CoV-2 in the hospital setting: a cohort study" (PMEDICINE-D-21-02091R2) for review by PLOS Medicine.

I have discussed the paper with my colleagues and the academic editor and it was also seen again by four reviewers. I am pleased to say that provided the remaining editorial and production issues are dealt with we are planning to accept the paper for publication in the journal.

[LINK]

We look forward to receiving the revised manuscript by Sep 10 2021 11:59PM.   

Sincerely,

Louise Gaynor-Brook, MBBS PhD

Associate Editor 

PLOS Medicine

plosmedicine.org

Requests from Editors:

Title: Please add ‘in the UK’ to your title. We suggest “Transmission of community- and hospital-acquired SARS-CoV-2 in hospital settings in the UK: a cohort study” 

Abstract:

Please structure your abstract using the PLOS Medicine headings (Background, Methods and Findings, Conclusions).

The final sentence of your Abstract Background should clearly state the study question e.g. “The aim of our study was to…” 

Abstract Methods and Findings:

Line 54 - please specify the incubation periods used to define community- or hospital-acquired cases

Please report percentages to at least one decimal place.

Please provide mean +/- SD for age

Line 61 - please report results on infections per 1000 susceptible patients per day to one decimal place

Line 62 - please add ‘per day’

Line 71 - Please move the final sentence earlier within your Abstract Methods and Findings

Abstract Conclusions: line 75 - please be careful to avoid inadvertent use of causative language - please revise “pose a substantial infection risk” to speak instead of associations

Thank you for providing an Author Summary. Please revise ‘nosocomial’ on line 89, which may not be a familiar term to a non-technical audience, and temper inadvertent causative language on line 110 to speak instead of associations. In the final bullet point of ‘What Do These Findings Mean?’, please describe the main limitations of the study in non-technical language.

Methods:

Please state early in the Methods section that your study did not have a prospective protocol or analysis plan, alongside your statement that ‘Analyses were data-driven’. We suggest adding this in the paragraph starting on line 185. 

Results: 

Please report percentages to at least one decimal place.

Discussion:

Line 453 - please revise to ‘ associated with more modest increases’

Line 526 - please be careful to avoid inadvertent use of causative language - please revise “pose a high risk” to speak instead of associations

Figures:

Figure 1 legend - sentence ending ‘consistent with COVID-19’ appears incomplete

Figures 4 & 5 - please clarify what is represented by the central dot within each horizontal bar 

Figure S7 - please amend typographical error in x axis label

Tables:

Tables 3 & 4 - please add sentences beginning “all independent variables...' and ‘The corresponding univariable’ as footnotes. Please make it clear in the table footnote that a value of 1 in the cells for aOR represents the comparison group for categorical variables (you may wish to denote this with an additional number in superscript)

Supplementary files: 

Please amend references contained in Figure S6 to adhere to PLOS Medicine style https://journals.plos.org/plosmedicine/s/submission-guidelines#loc-references

Comments from Reviewers:

Reviewer #1: Thank you for responding to my comments. I think the manuscript is improved and will be an excellent addition to the literature.

Reviewer #2: The authors have addressed all my points.

Michael Dewey

Reviewer #3: The authors are to be congratulated for their responsive revision. This paper very helpfully quantifies transmission risk for patients and HCWs and provides evidence that the greatest risk for transmission is from unsuspected cases rather than known Covid cases (e.g. hospital-acquired Covid in a patient, occult positive staff member) and that the greatest transmission risk is with *less* sick patients (i.e. non-ICU) rather than ICU patients with Covid. The major limitations remain the lack of case-by-case source analyses, lack of sequencing, and lack of CT values to quantify infection relative to viral burden but the big picture messages from the paper continue to ring true and are important to disseminate.

Reviewer #4: Thank you for addressing my comments!

[LINK]

---

## [Editor Report · Decision Letter 3]

14 Sep 2021

Dear Dr Yin, 

On behalf of my colleagues and the Academic Editor, Prof. Mirjam Kretzschmar, I am pleased to inform you that we have agreed to publish your manuscript "Transmission of community- and hospital-acquired SARS-CoV-2 in hospital settings in the UK: a cohort study" (PMEDICINE-D-21-02091R3) in PLOS Medicine.

PRESS

We frequently collaborate with press offices. If your institution or institutions have a press office, please notify them about your upcoming paper at this point, to enable them to help maximise its impact. If the press office is planning to promote your findings, we would be grateful if they could coordinate with medicinepress@plos.org. If you have not yet opted out of the early version process, we ask that you notify us immediately of any press plans so that we may do so on your behalf. Please also consider providing us with Twitter handle(s) that would be appropriate to tag (including your own, your coauthors’, your institution, funder, or lab) to assist with our efforts to promote your study.

Sincerely, 

Louise Gaynor-Brook, MBBS PhD 

Associate Editor 

PLOS Medicine